# Membrane Progesterone Receptors (mPRs, PAQRs): Review of Structural and Signaling Characteristics

**DOI:** 10.3390/cells11111785

**Published:** 2022-05-30

**Authors:** Peter Thomas

**Affiliations:** Marine Science Institute, The University of Texas at Austin, 750 Channel View Drive, Port Aransas, TX 78373, USA; peter.thomas@utexas.edu; Tel.: +1-361-749-6768

**Keywords:** mPRα, PAQR7, modeling, G-protein, ligand-binding domain, PGRMC1, 2nd messengers

## Abstract

The role of membrane progesterone receptors (mPRs), which belong to the progestin and adipoQ receptor (PAQR) family, in mediating rapid, nongenomic (non-classical) progestogen actions has been extensively studied since their identification 20 years ago. Although the mPRs have been implicated in progestogen regulation of numerous reproductive and non-reproductive functions in vertebrates, several critical aspects of their structure and signaling functions have been unresolved until recently and remain the subject of considerable debate. This paper briefly reviews recent developments in our understanding of the structure and functional characteristics of mPRs. The proposed membrane topology of mPRα, the structure of its ligand-binding site, and the binding affinities of steroids were predicted from homology modeling based on the structures of other PAQRs, adiponectin receptors, and confirmed by mutational analysis and ligand-binding assays. Extensive data demonstrating that mPR-dependent progestogen regulation of intracellular signaling through mPRs is mediated by activation of G proteins are reviewed. Close association of mPRα with progesterone membrane receptor component 1 (PGRMC1), its role as an adaptor protein to mediate cell-surface expression of mPRα and mPRα-dependent progestogen signaling has been demonstrated in several vertebrate models. In addition, evidence is presented that mPRs can regulate the activity of other hormone receptors.

## 1. Introduction

Membrane progesterone receptors (mPRs) were first identified in fish ovaries two decades ago [1]. mPRs are specific, membrane-associated, high-affinity receptors that mediate rapid progesterone activation of intracellular signaling (non-classical) pathways to induce cellular responses that are often nongenomic but may also ultimately result in alterations in gene transcription [1,2]. An important feature of mPR-dependent signaling is that it is transduced through the activation of G proteins [3]. However, mPRs are not members of the large GPCR superfamily and instead belong to the progestin and adipoQ receptor (PAQR) family, which has a different bacterial origin than that of GPCRs [3,4]. A recent phylogenetic analysis suggests the five vertebrate mPR subtypes, mPRα (PAQR7), mPRβ (PAQR8), mPRγ (PAQR5), mPRδ (PAQR6), and mPRε (PAQR9), expanded from an ancestral mPRγ in invertebrates with the mPRα gene being the most recent addition and only present in gnathostome (jawed) vertebrates [5]. The presence of five mPR subtypes in vertebrates, with both different and overlapping tissue distributions, ligand specificity, and G protein activation, has complicated the investigation of their physiological functions. The finding that mPRs are ubiquitously expressed with the nuclear progesterone receptor (PR) and another putative progestin receptor, progesterone receptor membrane component 1 (PGRMC1) in vertebrate tissues [1,4,6,7], has further complicated the investigation of the roles of these different receptors in mediating the reproductive and non-reproductive functions of progesterone. Nevertheless, significant progress has been made in our understanding of the functions of mPRs through the use of mPR-specific agonists and siRNA technology over the past two decades. The majority of this research has been conducted with mPRα, the dominant and likely the most physiologically relevant mPR subtype in reproductive tissues. For example, there is abundant evidence in several fish species that mPRα and mPRβ mediate progestogen hormone induction of oocyte meiotic maturation and that activation of mPRα induces sperm hypermotility and increases fertility [8,9]. Similar roles of mPRs in oocyte maturation have been identified in *Xenopus* and proposed in bovine oocytes [10,11]. Moreover, mPRα has been detected in human sperm, and there is an association between high expression of the receptor and sperm motility [12].

The widespread distribution of mPRs in reproductive tissues, their hormonal regulation, changes in mPR expression during the reproductive cycle, critical mPR functions in different reproductive tissues, and their potential involvement in reproductive tissue cancers have been the subjects of several recent extensive reviews [8,9,13,14,15,16,17,18], so they are not discussed further here. Nevertheless, several controversies over the structural characteristics, membrane topology, and signal transduction of mPRs have not been completely resolved [2,19,20]. Recently, a homology model of mPRα has provided new insights into its structure and membrane topology [21]. Moreover, the ligand-binding domain (LBD) of human mPRα has been identified through homology modeling, mutational analysis, and ligand specificity studies [21]. The identification of the structure of LBD of mPRα satisfies the remaining essential criterion for mPRα’s designation as a steroid receptor. In addition, they provide novel insights into the molecular mechanisms of ligand activation in mPRs, information necessary for developing new mPR-selective agonists and antagonists as well as determining the human health implications of mPR mutations.

Therefore, the present paper reviews the structural characteristics of mPRs, their ligand specificity, and signaling through activation of G proteins. In addition, the interactions of mPRs with PGRMC1 and other adaptor proteins, APPL1 (adaptor protein, phosphotyrosine interacting with pH domain and leucine zipper 10) and VLDL (very low-density lipoprotein receptor), are discussed. mPR regulation of PRs, γ-aminobutyric acid type A (GABA_A_) receptors, and G protein-coupled estrogen receptor (GPER) are also described.

## 2. Cellular Distribution and Membrane Topology

The mPR genes encode peptides with 330–377 residues (molecular weights~40 kDa) that form multiple transmembrane regions [4,22]. Although the majority of studies show mPR proteins are expressed on the plasma membranes of vertebrate cells, they are also often present intracellularly in the perinuclear region where the endoplasmic reticulum is located [1,2,3,8,19]. Localization of mPRs in this region is not unexpected since membrane receptors are trafficked to the cell membrane from the endoplasmic reticulum, which usually retains most of the receptor protein [23]. After ligand binding, mPRs are rapidly internalized by a clathrin-dependent mechanism [24,25], resulting in decreased cell-surface expression of the receptor, which is slowly restored [24]. The restoration of mPRs on the cell surface involves the participation of various adaptor proteins.

There has been considerable debate over the topology of mPRs in cell membranes. Initial computer hydrophilicity and structural analyses indicated that mPRs have seven transmembrane (TM) domains [4,22], and flow cytometry studies with antibodies directed towards peptides in their N- and C- terminal domains suggested that the N-terminal was extracellular and the C-terminal intracellular, similar to the orientation of GPCRs [3]. However, this orientation is opposite to that predicted for PAQRs and demonstrated for the adiponectin receptor (AdipoR) members of the family, AdipoR1 and AdipoR2 [4,26]. More recent structural analyses suggest the N-terminal of mPRs is intracellular [20,27], which is supported by the recent prediction of the “positive-inside rule” in which higher proportions of positively charged amino acids are intracellular [21]. This orientation is supported by results using *Xenopus* mPRβ clones with N- terminal fluorescent tags under permeabilized and nonpermeabilized conditions [25]. The C-terminal domain of mPRs is longer than that of the AdipoRs and is predicted to form an eighth TM domain with an intracellular C-terminal [20,27]. This is also supported by recent hydrophobicity and positive inside charge analyses [21]. Site-directed mutational analyses of the C-terminal domain of mPRα suggests it is involved in coupling to G proteins and their activation [3], which could explain why AdipoRs, which lack this region, are not able to activate G proteins.

## 3. Ligand Binding

A critical criterion for receptor designation is the presence of specific, high-affinity ligand binding. This criterion has been met by different laboratories for mPRs in prokaryotic (*Escherichia coli*), yeast, numerous mammalian expression systems, and most recently with purified recombinant mPRα coupled to graphene quantum dots [1,3,8,20,28]. Long-term culture of mammalian cells transfected with mPRs is required to obtain sufficient expression of the protein on the plasma membrane to detect progesterone binding [2]. Receptor activity has to be rapidly assayed within 30 min and at 4 °C in order to prevent mPR degradation and loss of progesterone binding [2,8], which may explain the failure of Brosens and colleagues to detect membrane localization and binding of the mPRs [19]. mPRα displays high affinity (Kd 2.5–7 nM), limited capacity (Bmax 0.03–0.72 nM), specific binding for the principal progestogen hormones in vertebrate species, including progesterone in mammals and other tetrapods, and hydroxylated progesterone derivatives in teleost species [2]. Among other natural steroids, only 21-hydroxyprogesterone and testosterone show significant binding to human mPRα with relative binding affinities (RBA) of approximately 20% that of progesterone [3].

Interestingly, the neurosteroid progesterone metabolite, allopregnanolone, also binds human mPRs, with RBAs of ~5% for mPRα and mPRβ and a three-fold higher affinity for mPRδ, which is the major neural mPR subtype and is expressed throughout the human brain [29]. mPRδ also shows a higher binding affinity than the other mPR subtypes for other neurosteroids such as dehydroepiandrosterone which has an RBA of 6.5%. Allopregnanolone and its 3-methylated synthetic analog, Ganaxolone, in addition to modulating GABA_-A_ receptor activity, also act as mPR agonists at low concentrations (20 nM) on neuronal cells to activate signaling pathways resulting in reductions in apoptosis and cell death [29,30].

The structure/binding and activity relationships of natural and synthetic steroids for mPRα have been investigated extensively [3,21,29,31,32]. The different ligand-binding characteristics of mPRs and PR have been exploited to distinguish progesterone actions through mPRs and PRs using a pharmacological approach. The potent nuclear PR agonist, promegestone (R5020), has a low RBA (4%) for mPRα [3] and does not activate mPRα at low nanomolar concentrations, whereas 10-ethenyl-19-norprogesterone (Org OD 02-0, 02-0) has a high RBA (>100%) for mPRα and is a potent mPR agonist, but shows no agonist activity through the PR [31]. Activation of mPRs in a variety of cell models, including fish sperm and oocytes, rodent Schwann cells, and human glioblastoma, lung, breast, and placenta cells, has been identified using low nanomolar concentrations of 02-0 [33,34,35,36,37,38,39]. A natural steroid, 17α-hydroxyprogesterone, has also been used to identify mPR-specific actions in human breast cancer cells, although high concentrations (1 µM) of this agonist are required to activate the mPRs [40], which is expected due to its relatively low RBA for mPRα (~1% that of progesterone) [3].

## 4. Structure of the Ligand-Binding Domain

The ligand-binding domain (LBD) of mPRα has recently been identified by Kelder and colleagues through a combination of homology modeling based on the known structure of another class of PAQRs, the AdipoRs, mutational analysis of critical amino acid residues in the binding pocket, and binding of progestogens and androgens with different functional groups [21]. The modeling predicted that glutamine 206 on TM 5 of the binding pocket has an essential H-bond interaction with the 20-carbonyl of progesterone. This was confirmed by mutational analysis with substitution at this position with alanine which is not able to donate an H-bond, resulting in a loss of progesterone binding. However, substitution with an arginine residue also resulted in no progesterone binding, which was unexpected because this amino acid is polar and can form H bonds.

X-ray analyses of AdipoRs indicate an oleic acid occupies the binding pocket [41], which suggests that mPRs may also have a free fatty acid in the binding pocket. Modeling predicted that the strong positive charge of the arginine 206 mutant would stabilize the binding of oleic acid to the mPRα binding pocket and that the addition of high concentrations of zinc would form a salt with the fatty acid in the binding pocket, permitting progesterone to bind [21]. This was supported by experiments showing that [^3^H]-progesterone binding to the arginine mutant was restored in the presence of 100 µM zinc. Interestingly, AdipoR1, which has an arginine residue in this region, was also able to bind progesterone when 100 µM zinc was added, indicating the similarities of the ligand-binding domains of the mPRs and AdipoRs. Consistent with this, the AdipoR synthetic agonist, AdipoRon, also has an affinity for mPRα and activates mPR-dependent G proteins and second messenger signaling [21].

Although the modeling predicted hydrophobic interactions between amino acids surrounding the binding pocket and progesterone, such as valine 146 in transmembrane 3, no hydrogen donor was identified in the vicinity of the 3-keto progesterone, which suggests this functional group is not required for progesterone binding to mPRα. This was confirmed with structure-binding activity experiments, which showed 3-deoxy steroids had similar RBAs to their corresponding 3-keto analogs [21]. Similarly, another research group has shown that several progesterone analogs, which lack oxygen at this position, have equal or higher RBAs than an analog with a 3-keto group [32]. Thus, the 3-keto is not required for binding to mPRα, whereas it is essential for progesterone binding to the LBD of the PR [42]. These, as well as other differences in ligand binding to mPRs and PRs [21,31,32,42], provide the basis for developing new mPR-selective ligands for studying mPR functions.

## 5. Signaling through G Proteins

### 5.1. G Protein Activation

Although evidence has been obtained since the earliest mPR publications that progestogen induction of intracellular signaling is mediated through activation of G proteins [1,3,8], their role in mPR signaling has been questioned [43] because mPRs are structurally unrelated to GPCRs and G proteins are not involved in adiponectin signaling of another class of PAQRs, the AdipoRs [44,45]. In addition, progestogens were shown to activate mPR signaling when they were coupled to a reporter in a yeast recombinant expression system lacking G proteins, which suggested that G proteins are not required for mPR signal transduction in the yeast model [20,43]. Although it has been claimed that mouse mPRβ signaling in PC12 neuronal cells is G protein-independent based on an experiment showing a lack of a cAMP response to progesterone treatment, the evidence is unconvincing because it was not accompanied by additional experimental approaches to investigate G-protein activation or signaling described below [46]. In contrast, there is extensive evidence that mPR-dependent progestogen intracellular signaling is mediated through G proteins in numerous vertebrate cell types, including fish oocytes, sperm, and ovarian follicle cells, human and bovine T lymphocytes, vascular endothelial and smooth muscle cells, breast cancer cell lines, and GnRH secreting and other neuronal cells [2,3,8,21,29,34,37,47,48,49,50,51,52,53,54,55]. Progestogen treatments increase [^35^S]GTPγS binding to the plasma membranes of mPR-expressing cells, which is indicative of G protein activation, and the identity of the activated G protein has been determined by immunoprecipitation of the radiolabeled GTPγS with specific antibodies of the α subunits of inhibitory and stimulatory G proteins [3,21,29,47,48,49,50,51,52]. Treatments with non-radiolabeled GTPγS, pertussis toxin, and cholera toxin cause G proteins to dissociate from their receptors and result in reductions in the number of receptor binding sites and ligand binding [53]. The finding that these treatments also decrease [^3^H]-progestogen binding to mPRs indicates they are closely associated with G proteins [3,50,51,52,53].

### 5.2. Association of mPRs with G Proteins

Co-immunoprecipitation studies and in situ proximity ligation assays (PLA) have provided additional evidence that all five mPRs are associated with G proteins. For example, Figure 1A shows a close association (<40 nm) between mPRα and an inhibitory G protein (G_i_) in human vascular smooth muscle cells in a PLA by the presence of red dots after pretreatment with inactivated pertussis toxin (iPTX). In contrast, a marked decrease in the number and staining of red spots is observed (Figure 1B) after treatment with activated pertussis toxin (aPTX), which decouples inhibitory G proteins from bound receptors [48,53]. PLA of mPRs overexpressed in several breast cancer cell lines with G proteins confirms previous results from co-immunoprecipitation and G protein inhibitor studies that indicate mPRα, mPRβ, and mPRγ are coupled to inhibitory G (G_i_) proteins, whereas mPRδ and mPRε are coupled to stimulatory G (G_s_) proteins [21]. One exception to this pattern is the coupling of mPRα to an olfactory stimulatory G (G_olf_) protein in sperm from two teleost species (51,53). Finally, there are no reports of activation of a G_q_ protein through mPRs, although progesterone has been shown to induce intracellular calcium mobilization in CHO cells transfected with ovine mPRα [56].

## 6. Second Messenger Signaling through G Proteins

### 6.1. cAMP/PKA Signaling

Upon activation of inhibitory G proteins (G_i_), the α subunit uncouples from the heterotrimeric G protein and downregulates membrane adenylyl cyclase (AC) activity resulting in decreases in intracellular cAMP levels and protein kinase A (PKA) activity, while activation of stimulatory G proteins (G_s_ and G_olf_) increases adenylyl cyclase activity and causes cAMP levels and PKA activity to increase. The cAMP responses to progestogen treatments mediated through mPRs are consistent with the G proteins they are coupled to in all the cell and tissue models investigated to date, thereby providing further confirmation that the mPRs signal through G proteins [2,3,8,9,21,29,33,34,37,46,48,49,51]. This is also supported by the finding that progestogen-induced changes in cAMP levels mediated through mPRs are attenuated by inhibitors of G protein activation such as pertussis toxin and cholera toxin [1,35,37,39,46,49,51,52]. These mPRα-induced changes in cAMP levels mediate critical functions such as oocyte meiotic maturation and sperm hypermotility in teleost fishes and relaxation of human vascular smooth muscle cells [9,13,49,52].

### 6.2. PI3K/AKT and MAPkinase/ERK1/2 Signaling

Progestogen hormone activation of mPRs also likely activates signaling pathways through βγ-subunit signaling [57], including the phosphatidylinositol 3-kinase (PI3K)/serine–threonine kinase (AKT) and mitogen-activated protein kinase (MAPkinase) pathways to mediate non-classical progestogen actions in numerous vertebrate cells, including fish oocytes, ovarian follicle cells, and sperm, as well as in human breast and lung cancer cells, neuronal cells, and vascular endothelial and smooth muscle cells [9,29,35,36,37,39,40,46,47,50,58,59,60]. For example, activation of these mPR-dependent signaling pathways inhibits apoptosis and cell death in fish ovarian follicle cells and human neuronal and breast cancer cells [29,37,50], which is not unexpected since PI3K/AKT and MAPkinase exert antiapoptotic actions and promote cell survival in numerous mammalian cells [61]. mPRα-dependent activation of the PI3K/AKT pathway increases phosphodiesterase (PDE) activity in full-grown teleost oocytes, decreasing cAMP levels and promoting oocyte maturation [12]. Although the same pathway is activated in croaker sperm to induce sperm motility, the mechanism through which increased PDE activity increases sperm motility remains unclear. Activation of PI3K/AKT signaling through mPRα induces either pro- or anti-tumorigenic responses in breast and other cancer cells. For instance, whereas mPRα-mediates reversal of epithelial to mesenchymal transition (EMT) through PI3K and EGFR pathways in MDA-MB-468 breast cancer cells [62], mPRα-dependent activation of PI3K/AKT/mTOR signaling in MDA-MB-453 breast cancer cells transfected with breast cancer resistance protein (BCRP) resulted in increased BCRP expression which would promote breast cancer metastasis [40].

### 6.3. Activation of Multiple Signaling Pathways

Activation of PI3K/Akt signaling through mPRs is usually accompanied by induction of MAPkinase signaling and ERK1/2 phosphorylation [8,9,25,37,47,48,50,51,60,63]. Both PI3K/Akt and MAPkinase signaling pathways mediate the functions of mPRα in various cell models, including progestogen stimulation of sperm motility and oocyte maturation in fish [8,9], anti-apoptosis in fish ovarian follicle and human breast cancer cells [37,51,60], nitric oxide production in human umbilical vascular endothelial cells [48], and the progesterone-induced increase in sarcoplasmic reticulum calcium levels and decrease in myosin light chain (MLC) phosphorylation in human vascular smooth muscle cells resulting in their relaxation [50,63]. Interestingly, progesterone activation of mPRα has an opposite effect on MLC phosphorylation in human myometrial cells collected at the end of pregnancy, increasing it through p38MAPK, which enables the myometrium to contract [47]. PI3K and EGFR are intermediaries in the activation of ERK1/2 [57,64]. Progestogen induction of ERK1/2 phosphorylation in breast cancer cells stably transfected with seatrout mPRα is dependent on EGFR transactivation [59]. Teleost sperm hypermotility through mPRα is also dependent on EGFR transactivation, which suggests that ERK1/2 activation in sperm may also be mediated through EGFR [65]. mPRα activation also results in activation or inhibition of EGFR signaling in lung and breast cancer cells [39,62]. In addition, a variety of downstream intracellular mediators of mPR signaling have been identified, including JNK, mTOR, NFκB, Snail, and CREB [13,40,66,67,68]. Thus, a variety of signaling pathways, including adenylyl cyclase/cAMP/PKA, PI3k/AKT, MAPkinase/ERK1/2/PDE, and EGFR pathways, are mediated through mPR-dependent G protein activation, often in the same vertebrate cells. For example, all these signaling pathways are involved in the upregulation of fish sperm motility, suggesting the presence of a complex signaling system regulating this critical function of sperm [9].

## 7. Association of mPRs with Adaptor Proteins

### 7.1. Association of mPRα with PGRMC1

Numerous and diverse functions have been reported or proposed for PGRMC1 [2,6,69,70], including binding progesterone and other steroids and its association with and trafficking or stabilization of a diverse array of molecules, including heme, cytochromes P450, EGFR, and the insulin receptor [71,72,73,74], which suggests it can function as an adaptor and chaperone protein. The finding that PGRMC1 associates with such a wide array of compounds complicates the identification of its true functions distinct from those of the molecules associated with it. The possible association of PGRMC1 with mPRα was investigated since both proteins are intermediaries in progesterone actions and may be components of a progesterone membrane receptor complex [75]. Transfection of PGRMC1 mRNA into MDA-MB-231 cells, which display weak plasma membrane expression of mPRα and low [^3^H]-progesterone membrane binding, caused upregulation of both PGRMC1 and mPRα proteins on the cell membrane and increased progesterone membrane receptor binding activity characteristic of mPRα, with a Kd of ~5 nM and high-affinity binding for the mPR agonist, 02-0 [75]. Progesterone treatment caused G protein activation in PGRMC1-transfected cells, which was blocked by transfection with mPRα siRNA, and this treatment also reduced [^3^H]-progesterone binding and abrogated the antiapoptotic actions of progesterone. Co-immunoprecipitation and immunocytochemistry experiments indicated a close association of mPRα and PGRMC1 in several PR-negative breast cancer cell lines [75]. These results provided the first evidence that PGRMC1 can act as an adaptor protein for mPRα, as has been proposed for PGRMC1 with other molecules [74]. Furthermore, it suggests that the two proteins may act as a receptor complex, both being required to mediate non-classical progesterone signaling in cells [75].

A subsequent study in human granulosa/luteal cells showed that progesterone suppression of entry into the cell cycle was dependent on the expression of mPRa, PGRMC1, and PGRMC2, which were shown to be closely associated with PLA, which led to the authors’ suggestion that they form a complex required for progesterone signaling [76]. A close association between PGRMC1 and mPRα, PGRMC1-dependent upregulation mPRα protein expression on the cell membrane, and PGRMC1 regulation of mPRα functions has also been demonstrated in zebrafish oocytes [34]. Microinjection of PGRMC1 antisense morpholinos into zebrafish oocytes downregulated both PGRMC1 and mPRα protein expression on the oocyte membrane and blocked progestogen induction of oocyte meiotic maturation [34], which previous studies had shown is mediated through mPRα [59]. Moreover, in situ PLA showed a close association between PGRMC1 and mPRα on oocyte membranes by the presence of red dots (Figure 2A, PG-mPRα), whereas red dots were absent in the negative control in which the mPRα antibody was replaced with IgG (Figure 2B, PG-IgG) [34].

Further support for an interaction between PGRMC1 and mPRα was obtained in zebrafish ovaries in which PGRMC1 was globally knocked down by CRISPR/Cas9 gene editing [77]. Progestogen induction of oocyte maturation was decreased in the PGRMC1 knockouts, which was associated with decreased protein expression of mPRα [77]. The results from these four studies suggest that PGRMC1 is closely associated and forms a receptor complex with mPRα necessary for mediating mPRα-dependent and PGRMC1-dependent progesterone signaling and functions in different vertebrate cells. Recent crystallographic analysis of the cytosolic domain of PGRMC1 shows it exists as a dimer via interactions of its two haem molecules [78]. This haem stacking dimer of PGRMC1 is dissociated by the gas mediator, carbon monoxide [79]. Dimerization of PGRMC1 through haem-haem stacking interactions is required for interactions with EGFR and cytochromes P450 [78] and may also be necessary for interactions with mPRα, although this remains to be investigated. Both PGRMC1 and mPRα are broadly distributed in vertebrate tissues and could potentially interact to mediate PR-independent progesterone responses in a wide variety of cells [4,6]. Therefore, future investigations on non-classical progesterone signaling should include an assessment of the roles of both mPRα and PGRMC1 when they are detected together in a cell or tissue model.

### 7.2. Association of mPRβ with APPL1 and VLDL

Information on potential interactions between PGRMC1 and other mPR subtypes in vertebrate cells is currently lacking. mPRβ mediates progesterone-induced meiotic maturation of *Xenopus* oocytes [10,25] and also participates in progestin-induced maturation of zebrafish oocytes [80]. Nader and colleagues have provided evidence that APPL1 interacts with mPRβ in *Xenopus* oocytes and that this interaction is essential for progesterone induction of oocyte maturation [25]. These authors suggested that APPL1 is likely involved in progesterone-induced endocytosis of mPRβ [25]. APPL1 is an adaptor protein that binds to AdipoRs and has an indispensable role in adiponectin signaling [81], so it is not surprising that APPL1 may act as an adaptor protein with another PAQR, mPRβ. Nader and coworkers also reported that the VLDL receptor acts as a chaperone for *Xenopus* mPRβ trafficking to the plasma membrane as well as from the endoplasmic reticulum to the Golgi, and VLDL is also required for progesterone signaling through the receptor [82]. In conclusion, there is clear evidence for mPR interactions with these proteins that can act as mPR chaperones and adaptor proteins necessary for non-classical progesterone signaling. However, there are many unanswered questions, such as do PGRMC1, APPL1, and the VLDL receptor interact with multiple mPR subtypes, and do they act in concert to form large receptor complexes to mediate progesterone signaling?

## 8. mPR Regulation of PRs, GABA_A_ Receptors, and GPER

### 8.1. Regulation of PR Transactivation

Karteris and coworkers obtained the first evidence that mPRs can regulate the activity of other progesterone receptors [9,47]. These investigators examined mPR regulation of the PR by measuring the binding of the PR to a glucocorticoid-response element coupled to a luciferase reporter vector transiently transfected into human myometrial cells with high expression of PR-B, representing the myometrium at an early stage of pregnancy [47]. Treatment with siRNAs for mPRα and mPRβ and with pertussis toxin markedly attenuated the progesterone-induced increase in luciferase activity, which suggests that progesterone acts through mPRα and mPRβ Gi-dependent pathways in addition to PR_B_ to transactivate the PR during this stage of pregnancy [9,47]. In contrast, progesterone-BSA treatment of myometrial cells collected at the termination of pregnancy when mPR levels are elevated caused a decrease in steroid receptor co-activator SRC-2 mRNA expression, which was blocked by treatment with mPRα siRNA, which suggests that mPRα can also down-regulate transactivation of PR in the myometrium during labor [47]. On the basis of these and other results, Karteris et al., proposed that activation of mPRs amplifies the action of PR_B_ to maintain the myometrium in a quiescent state during early pregnancy and at the end of pregnancy contributes to a functional progesterone withdrawal allowing the myometrium to become contractile.

### 8.2. Regulation of GABA_A_ Receptor Phosphorylation

A recent study indicates that mPRs can also influence the activity of another class of receptors, GABA_A_ receptors, that are allosterically modulated by progesterone and its metabolite allopregnanolone [83]. These neurosteroids are also mPR agonists [29,30]. Treatment of mice hippocampal slices with the specific mPR agonist 02-0 mimicked the effects of allopregnanolone to increase phosphorylation of Ser-408 and Ser-409 in the GABA_A_ receptor beta 3 subunit through cAMP-dependent PKA and protein kinase C (PKC) pathways [83]. These treatments also increased the expression of the GABA_A_ receptor on the plasma membranes of hippocampal cells. While these actions of allopregnanolone correlate with enhanced continued GABAergic inhibition, mPR activation did not directly allosterically modify GABA_A_ receptor activity in HEK293 cells expressing the GABA_A_ receptor and instead caused a sustained elevation of tonic current [83]. Interestingly, this effect of 02-0 was blocked by PKA and PKC inhibitors and by impairment of SER-408/9 phosphorylation, which also blocked the effects of sustained allopregnanolone exposure on tonic inhibition. The authors concluded that mPR-dependent GABA_A_ receptor phosphorylation mediates these metabotropic effects of allopregnanolone and probably other neurosteroids [83].

### 8.3. Regulation of GPER Expression

Reciprocal regulation of mPRs and GPER has been demonstrated in zebrafish oocytes by specific agonists for these receptors to control the onset of oocyte meiotic maturation [8,84]. Estrogens produced in ovarian follicle cells act via GPER coupled to a G_s_ to increase cAMP levels in growing oocytes and maintain meiotic arrest, whereas fish progestogens produced by follicle cells in response to gonadotropin stimulation induce maturation of full-grown oocytes by decreasing cAMP levels through a mPRα/G_i_-dependent signaling pathway [1,8,84]. Four-hour treatments with a fish progestogen hormone and 02-0 decreased GPER expression and upregulated mPRα expression in oocyte membranes, whereas treatments with estradiol-17β and the GPER agonist, G-1, had opposite effects, decreasing mPRα expression and increasing that of GPER [8,84]. This reciprocal hormonal regulation of mPRs and GPER is proposed to be a critical component of the dual control by these receptors of the onset of oocyte maturation in teleost fishes [8,84].

## 9. Conclusions and Future Studies

Recent significant progress has been made in determining the membrane topology and structure of the LBD of mPRs through homology modeling and mutagenesis. Experimental confirmation of the structures of mPRs, such as by X-ray analysis of their three-dimensional structures, will be required to verify the predictions of the homology model. A specific mPR antagonist has not been identified to date. New information on the structural requirements for binding of ligands to mPRs is enabling the development of mPR-specific ligands, including those that may act as antagonists, that can be used to investigate the functions of mPRs [21,31,32,85]. It is now evident that mPRs are closely associated with G proteins and adaptor proteins such as PGRMC1. However, direct evidence that mPRs physically interact with these proteins is currently lacking. A combination of biophysical approaches such as nuclear magnetic resonance (NMR) and surface plasmon resonance, together with biochemical methods, for example, bioluminescence resonance energy transfer (BRET) and mass-spectroscopy of the affinity-purified proteins, will be necessary to confirm these proposed protein-protein interactions [86,87,88,89]. Moreover, detailed knowledge of the structures of mPRs will be required in order to determine the molecular mechanisms controlling their interactions with these proteins. This information will be essential for understanding the factors that influence these interactions and mPR functions in health and disease. Twenty years after their discovery, many new and interesting features of mPRs are being identified. The model in Figure 3 summarizes current knowledge of the G proteins and major second messenger pathways activated by mPRα, coupling of mPRα to PGRMC1 on the cell membrane, and evidence for regulation of other receptors by mPRα.

## Figures and Tables

**Figure 1 cells-11-01785-f001:**
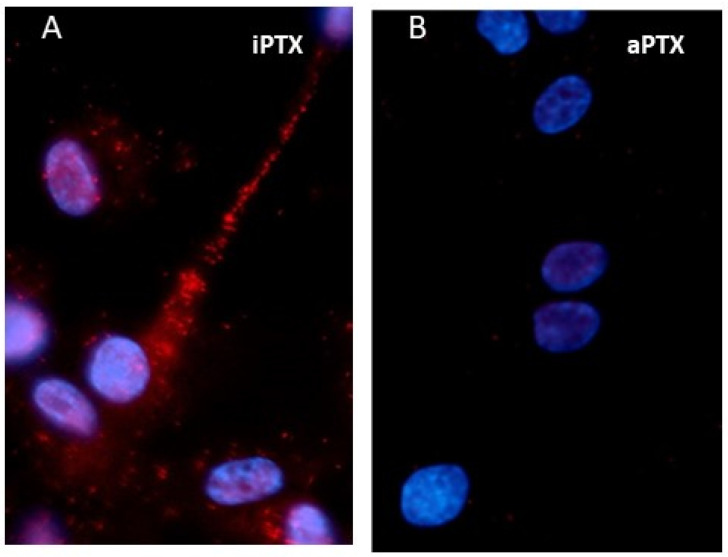
In situ proximity ligation analysis (PLA) of the interactions of mPRα with an inhibitory G protein (G_i_) using specific mPRα and G_i_α-subunit antibodies in human vascular smooth muscle cells (VSMCs). (**A**) A close association (distance < 40 nm) between mPRα and G_i_ is shown by the presence of red dots in the image, which had been preincubated with inactivated pertussis toxin (iPTX). (**B**) The PLA image shows a marked decrease in the number of red dots in VSMCs that had been pretreated with activated pertussis toxin (aPTX). Treatment with aPTX uncouples G_i_ from hormone receptors, whereas iPTX is inactive. Nuclei are stained blue with DAPI. Reproduced from Pang and Thomas [48] with permission.

**Figure 2 cells-11-01785-f002:**
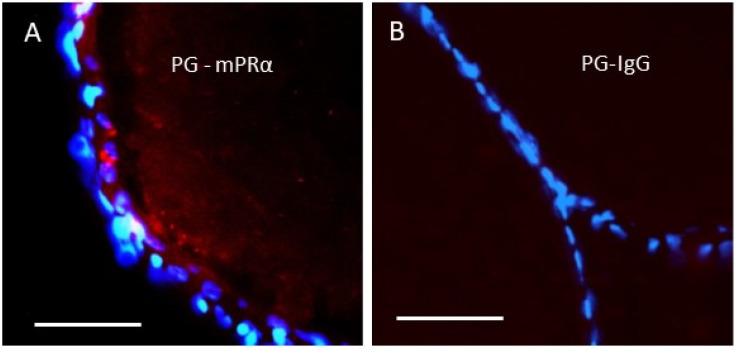
In situ proximity ligation analysis (PLA) of the association of mPRα with a PGRMC1 in zebrafish oocytes using zebrafish mPRα and PGRMC1 (PG) antibodies. (**A**) Close proximity of mPRα and PGRMC1 (<40 nm) in the image are shown as red dots. (**B**) PLA using the PGRMC1 antibody and IgG as a negative control showing the absence of red dots in the image. The nuclei (blue) of the follicle cells surrounding the oocyte are stained with DAPI. Scale bar 100 µm. Produced from Aizen et al. [34] with permission.

**Figure 3 cells-11-01785-f003:**
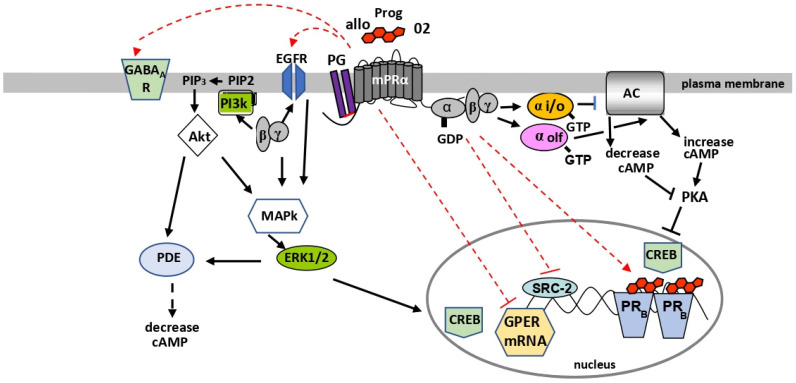
Current model of mPRα association with adaptor and G proteins, regulation of second messenger pathways (black), and regulation of other receptors (red). Hatched lines indicate intermediates/mechanisms of receptor regulation unknown. PG-PGRMC1—allo-allopregnanolone.

## Data Availability

Not applicable.

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
