# Peer review of "Membrane Progesterone Receptors (mPRs, PAQRs): Review of Structural and Signaling Characteristics"

_cells, 2022, doi:10.3390/cells11111785_

Round 1
Reviewer 1 Report
The manuscript by Peter Thomas titled “Membrane progesterone receptors (mPRs, PAQRs): Review of structural and signaling characteristics,” provides a clear and comprehensive summary of our current understanding the role progestin and adipoQ receptor (PAQR) family members play in mediating rapid, nongenomic progestogen actions in various cells types and tissues. The author does a good job of pointing out current controversies and debates regarding mPR mechanism of action and how these family members integrate into different signaling pathways. I view this manuscript as being a valuable resource for investigators that study rapid (i.e., membrane, nonclassical PGR) progestogen action. In general, the paper is well written and easy to read. The following are edits and comments provided to improve the clarity of the manuscript.
The author presents images and data from proximity ligation analysis (PLA) studies to support the association of mPRs with other cellular proteins (Sections 5 and 7). While PLA demonstrates that proteins of interest are in close proximity, it does not prove protein-protein interactions. It would be helpful for the reader to understand what future studies are needed to firmly establish mPR physically interacts with other proteins such as G-proteins and PGRMC1 (surface plasmon resonance, crystallography, etc.). In other words, what are the current gaps, and what studies are needed to firmly establish the role of mPRs in specific cell signaling pathways?
Minor points:
Line 68: The following sentence is a bit cumbersome “Therefore, the present paper will review the structural characteristics of mPRs, their ligand specificity and signaling through activation of G proteins, their interactions with PGRMC1 and other adaptor proteins, APPL1 (adaptor protein, phosphotyrosine interacting with pH domain and leucine zipper 10, VLDL (very low density lipoprotein receptor), and mPR regulation of PRs, γ-aminobutyric acid type A (GABAA), receptors, and G protein-coupled estrogen receptor (GPER). Perhaps break into two parts- potential mPR interactions, and its effects on other receptor systems.
Line 70: Missing end parenthesis for (adaptor protein, phosphotyrosine interacting with pH domain and leucine zipper 10)
Line 81: I recommend changing the term “Upon progesterone activation” to After ligand binding. The meaning of “progesterone activation” is a bit unclear.
Line 114: Do the authors mean “hydroxylated progesterone derivatives found in teleost species” instead of “and progesterone derivatives with multiple OH groups on the side chain in teleost species”?
Line 204: A comma should be inserted such that the sentence reads “pertussis toxin (aPTX), which decouples…”
Line 242: The sentence “For example, activation of these mPR-dependent signaling pathways inhibits apoptosis and cell death in fish ovarian follicle cells, and human neuronal, and breast cancer cells which is not unexpected since PI3K/AKT and MAPkinase exert antiapoptotic actions and promote cell survival in numerous mammalian cells.” should be changed to “For example, activation of these mPR-dependent signaling pathways inhibits apoptosis and cell death in fish ovarian follicle cells, as well as human neuronal and breast cancer cells, which is not unexpected since PI3K/AKT and MAPkinase exert antiapoptotic actions and promote cell survival in numerous mammalian cells.”
Line 291: Insert commas- “into MDA-MB-231 cells, which display weak plasma membrane expression of mPRα and low [3H]-progesterone membrane binding, caused..”
Line 305: Insert commas- “expression of mPRa, PGRMC1, and PGRMC2, which were shown..”
Line 354: Change to “with these proteins that can act”
Line 371- Change “suggests that mPRα can also down-regulates transactivation of PR” to “suggests that mPRα can also down-regulate transactivation of PR”
Line 388- “02” should be changed to “02-0” for consistency.
Author Response
I am very pleased you considered this review a valuable resource for investigators studying nonclassical progestogen actions. Thank you for your helpful review. I agree with your suggestion to include a discussion on the need for additional methods to confirm that mPRs physically interact with G proteins and adaptor proteins and have added several sentences (lines 419-425) and 4 references in the "Conclusions" section of the manuscript to address this. I stated that PLA showed a close association between mPRα with G proteins and PGRMC1 throughout the text with one exception (line 204) where I wrote that Co-IP studies and PLA " confirmed all five mPRs are coupled to G proteins" This has now been changed to " provide additional evidence that all five mPRs are associated with G proteins" . All the other minor points have been addressed in the revised manuscript, including revising the last sentence in the Introduction which is now three separate sentences.
Reviewer 2 Report
This is a comprehensive and exciting review on membrane progesterone receptors. My comments are minor.
line 50: "...........and that activation of mPRα......." Please insert the word "of".
line 83-84: the statement is unclear; it seems like two thoughts were being summarized here but not linked together coherently.
line 160: should be "also has affinity".
line 236-280: This is a very long paragraph and difficult to read through as is. Please consider restructuring as smaller sections.
A similar comment is that many other sections are very long and can benefit from being structured as sub-sections.
line 399: not sure what ".......02-0 down-regulated..." means. Can this be a typo?
An abbreviation list is missing and is required.
Author Response
I am very pleased you found my review on membrane progesterone receptors to be comprehensive and exciting. I agree with your comment that the paragraph on second messenger signaling (lines 236-281) is very long. It has now been divided into several paragraphs. Other long paragraphs in the manuscript have also been divided into multiple paragraphs. In addition, several sections of the manuscript now have subheadings : 5. Signaling through G proteins, 6. Second messenger signaling through G proteins, 7. Association of mPRs with adaptor proteins, 8. mPR regulation of PRs, GABAA receptors, and GPER. I think these changes have improved the readability of the manuscript. Other minor suggestions including the inclusion of an abbreviation list have been addressed.